# Galectin-9 as a biomarker for disease activity in systemic lupus erythematosus

**Naoki Matsuoka[1], Yuya Fujita[1], Jumpei Temmoku[1], Makiko Yashiro Furuya[1], Tomoyuki Asano[1], Shuzo Sato[1], Haruki Matsumoto[1], Hiroko Kobayashi[1], Hiroshi Watanabe[1], Eiji Suzuki[1], Hideko Kozuru[2], Hiroshi Yastuhashi[2], Kiyoshi Migita[1,2]***

**1** Department of Rheumatology, Fukushima Medical University School of Medicine, Fukushima, Japan,
**2** Clinical Research Center, Nagasaki Medical Center, Nagasaki, Japan

* migita@fmu.ac.jp

**Data Availability Statement:** All relevant data are within the manuscript and its Supporting Information files.

**Funding:** The study was supported by the Practical Research Project for Rare / Intractable Diseases

## Abstract

### Background

Systemic lupus erythematosus (SLE) is an autoimmune disease characterized by elevated interferon (IFN) signature genes. Galectin-9 (Gal-9) is a β-galactoside-binding lectin that is reportedly useful as a biomarker for IFN gene signatures. In a cross-sectional study of Japanese patients with recent-onset SLE, we aimed to determine whether raised serum Gal-9 levels were associated with the disease activity or organ damage seen in SLE patients.

### Methods

The current study included 58 Japanese patients with SLE and 31 age-matched healthy individuals. Disease activity and organ damage were assessed using SLE Disease Activity 2000 (SLEDAI-2K) and Systemic Lupus International Collaborating Clinics (SLICC) damage index. Serum and cerebrospinal fluid (CSF) Gal-9 concentrations were quantified using ELISA. Correlation analyses between Gal-9 and clinical parameters including disease activity were performed.

### Results

Serum levels of Gal-9 were significantly increased in patients with SLE compared with the control group (16.6 ng/ml, [interquartile range (IQR); 3.6–59.7] versus 4.74 ng/ml, [IQR; 3.0–9.5], $p$<0.0001). Gal-9 was significantly correlated with disease activity measures in the SLEDAI-2K. Serum Gal-9 levels were significantly greater in patients with SLE-related organ involvement (23.1 ng/ml, [IQR; 5.1–59.7] versus 12.5ng/ml, [IQR; 3.6–39.0], $p$ = 0.013). Whereas there was no difference in serum levels of CXCL10 or M2BPGi between patients with and without SLE-related organ involvement. Serum levels of Gal-9 were significantly higher in SLE patients with active renal involvement determined by BILAG renal score (A-B) compared to those without active renal involvement (C-E). Whereas there was no significant difference in serum levels of Gal-9 between SLE patients with or without active other organ involvements (neurological or hematological) determined by BILAG score. SLE

from Japan Agency for Medical Research and Development, AMED.

**Competing interests:** The study was supported by the Practical Research Project for Rare / Intractable Diseases from Japan Agency for Medical Research and Development, AMED. This does not alter our adherence to PLOS ONE policies on sharing data and materials.

patients with detectable circulating IFN-α had raised serum Gal-9 levels. Levels of Gal-9 were significantly higher in the CSF from patients with recent-onset neuropsychiatric SLE (NPSLE) than in those from non-SLE controls (3.5 ng/ml, [IQR; 1.0–27.2] versus 1.2 ng/ml, [IQR; 0.9–2.1], $p$ = 0.009).

## Conclusions

Gal-9 could be a serologic marker of disease activity and organ involvement in SLE patients. Future studies evaluating the role of Gal-9 in the SLE phenotype may provide insights into SLE pathogenesis.

## Background

Systemic lupus erythematosus (SLE) is a systemic autoimmune disease characterized by the loss of immunological tolerance against nuclear antigens [1]. The clinical and paraclinical tools to assess disease activity and predict the disease course are inadequate, and identification of easily accessible biomarkers is required for SLE [2]. Activation of the type I interferon (IFN) system is involved in the pathogenesis of SLE [3]. Therefore, type I IFN signatures, such as raised circulating levels of IFN-α or IFN-inducible genes could be linked with the disease activity and disease flares in SLE patients [4]. Surrogate markers for the IFN signature, such as CXCL 10, have been evaluated in SLE patients [5]; however, easy and accurate methods to measure IFN signatures have not been generally established [6]. More recently, Hoogenet et al. demonstrated that galactin-9 (Gal-9) is a novel, easy to measure biomarker for type1 IFN signatures and Gal-9 could aid in clinical decision marking in SLE [7]. Gal-9, one of the β-galactoside binding lectins, plays important regulatory roles in autoimmune diseases [8]. T cell immunoglobulin and mucin domain containing molecule-3 (Tim-3) expressed on T cells is involved in the regulation of Th1 cell-mediated immunity and has been identified as the ligand of Gal-9 [9]. Recent studies also suggest that Gal-9 can suppress the differentiation of Th17 cells in Tim-3-dependent or independent manners [10]. Due to the heterogeneity of the SLE disease phenotype, reliable biomarkers that reflect SLE disease activity and/or organ damage are required. Complement proteins or autoantibodies, such as anti-ds-DNA antibody, are used to monitor global disease activity [11]. However, these parameters could be associated with disease activity and may not reflect the SLE disease phenotype or associated organ damage [12]. Gal-9, which is a type1 IFN signature, should be further evaluated in SLE patients with various disease phenotypes. Mac-2 Binding Protein Gylcan Isomer (M2BPGi), which interacts with galectins, is a reliable marker for assessing liver fibrosis in autoimmune liver diseases [13]. The impact of M2BPGi on outcome was also demonstrated in SLE in addition to autoimmune liver diseases [14]. In this study we sought to determine the role of these circulating soluble proteins related to IFN signatures, including Gal-9, in patients with SLE with different levels of disease activity and disease phenotypes. We also examined the relationship of Gal-9 with disease activity and whether it is a useful biomarker for predicting disease activity including organ involvement in patients with SLE.

## Methods

### Patients and clinical evaluations

A total of 58 Japanese patients with recent-onset SLE were included in the study. SLE patients were enrolled within 32 months (mean 18 month, range 0–32) of SLE diagnosis, which was

based on the fulfillment the American College of Rheumatology (ACR) 1997 criteria [15]. All patients were treated in Department of Rheumatology, Fukushima Medical School from June 2009 to March 2019. All patients with SLE underwent a structured interview, physical examination, laboratory tests, and a review of medical records. In patients with SLE, disease activity and organ damage were ascertained with the Systemic Lupus Erythematosus Disease Activity Index (SLEDAI) [16] and the Systemic Lupus International Collaborating Clinics (SLICC) damage index [17], respectively. SLEDAI scores were recorded at the time of follow-up for SLE patients. SLE disease activity was also determined using the British Isles lupus assessment Group (BILAG) score which consisted of evaluation of 8 domains, general, musculocutaneous, neurological, musculoskeletal, cardio-respiratory, renal manifestations, vasculitis and hematological findings [18]. It was designed to reflect physicians' intention-to-treat with five categories (A, B, C, D and E). As a control group, 31 age- and sex-matched healthy controls (HCs; 5 males and 26 females, median age 39 years [26–52]) were enrolled. This study was conducted in accordance with the principles of the Declaration of Helsinki. Ethical approval for this study (No. 30285) was provided by the Ethics Committee of Fukushima Medical University and written informed consent was obtained from each individual.

## Serological analysis

Serum levels of complement 3 (C3) and serum complement 4 (C4), the presence of double strand (ds)-DNA and anti-nuclear antibodies (ANA), and the total number of white blood cells (WBCs) were measured in the clinical laboratory of Fukushima Medical University. Serum samples were obtained from 58 patients with SLE (50 women and 8 men; mean ± SD age 35.8 ± 1.8 years). Among these 58 patients, 5 patients were selected for consecutive analysis with serial serum samples drawn. These patients were chosen due to fluctuations in disease activity during the treatments.

## Cerebrospinal fluid (CSF)

CSF samples were obtained in all 18 patients with NPSLE (3 males and 15 females). Among these, 7 patients with NPSLE were included outside of the enrolled 58 SLE paints in whom serum samples were not collected. Their average age at the onset of CNS manifestations was 34.6 years (range 16–63 years). According to the ACR nomenclature [19], the symptoms of NPSLE exhibited by our patients were follows: aseptic meningitis (8), seizures (3), cognitive dysfunction (3), headache (1), cerebrovascular disease (1), myelopathy (1), cranial neuropathy (1).

For ethical reasons, CSF samples were not collected from SLE patients without any neuropsychiatric involvement or from healthy volunteers. Because of the difficulty in confirming neurologic diagnoses and of assigning cause to SLE, we defined NPSLE as the presence of at least 1 clinical feature of neuropsychiatric syndromes and at least 1 of the following: abnormal findings on brain magnetic resonance imaging, diffuse abnormal signal of brain single-photon-emission computerized tomography, severely abnormal results on a neuropsychiatric test, and elevated CSF IgG index or increased interleukin-6 (IL-6) activity in their CSF [20]. We also tested CSF samples from 6 patients with a history of headache who did not have SLE or any other autoimmune diseases, as non-SLE controls. Serum and CSF samples were prepared and stored at -70˚C until analyzed. All assays were performed without information of the diagnosis and clinical manifestations.

### Enzyme-linked immunosorbent assay for CXCL-10 and Galectin-9

Serum concentrations of Galectin-9 and CXCL10 were measured using human enzyme-linked immunosorbent assay kit (R&D Systems, Minneapolis, MN, USA) according to the manufacturer's instruction. The measurement of IFN-α was performed with the VeriKine ™ Human IFN-α ELISA Kit (Product #41100), following the manufacturer's instructions.

### Measurement of M2BPGi

Serum M2BPGi level was directly measured with the HISCL™ M2BPGi™ reagent kit (Sysmex, Kobe, Japan) using an automatic immunoanalyzer HISCL-5000 (Sysmex, Hyogo, Japan). M2BPGi levels were indexed using the following equation: Cut-off Index (C.O.I.) = ([M2BPGi]sample-[M2BPGi]NC)/([M2BPGi]PC)-[M2BPGi]NC), where [M2BPGi]sample represents the M2BPGi count of the serum sample (PC: positive control, NC: negative control) [21].

### Statistical analysis

Results were non-normally distributed and are presented throughout the manuscript with median and 25–75th centiles [median, IQR] and were compared by the Mann-Whitney U test. Correlations between continuous variables were analyzed by the Spearman's rank correlation test. Paired data were analyzed by non-parametric tests using the Wilcoxon signed-rank test for the comparison of paired data.

## Results

### Demographic and clinical characteristics in patients with SLE

We studied 58 patients with SLE and 31 healthy control subjects and the demographics and clinical characteristics of SLE patients are presented in Table 1. Patients were predominantly female (86%), and all patients were of Japanese ethnicity. Approximately 95% of patients had active disease (SLEDAI-2K > 4) and approximately 57% had any organ damage (SDI $\geq$ 1). Patients and HCs showed no significant differences in age or sex distribution.

### Correlation between circulating levels of Gal-9 and SLE disease activity

We measured serum levels of Gal-9 using specific ELISA assays. As shown in Fig 1, the serum levels of Gal-9 were significantly higher in patients with SLE compared with the healthy controls (16.6 ng/ml [IQR; 3.6–59.7] versus 4.74 ng/ml [IQR; 3.0–9.5], $p < 0.0001$).

Correlations between Gal-9 levels and disease activity and clinical serological parameters were examined. As sown in Fig 2A–2D, the serum levels of Gal-9 in SLE patients showed a moderate positive correlation with SLE disease activity as measured by the SLEDAI-2K (Fig 2A, $p < 0.001$, r = 0.47 [95%CI; 0.27–0.67]). Additionally, serum levels of Gal-9 showed a weak correlation with serum ds-DNA antibody titer (Fig 2B, $p = 0.02$, r = 0.30 [95%CI; 0.15–0.42]) and a negative correlation with serum levels of C3 (Fig 2C $p = 0.01$, r = −0.34 [95%CI; -0.54--0.11]). There was no significant correlation between serum levels of Gal-9 and C4 (Fig 2D, $p = 0.19$, $r = -0.18$ [95%CI; -0.32- -0.02]).

### Relationships between serum levels of biomarkers and organ involvements

CXCL-10 has been found to be elevated in SLE patients with high type I IFN gene signatures [5]. Therefore, we measured CXCL 10 and M2BPGi using the same sera and tested their

**Table 1. Baseline characteristics of 58 Japanese patients with SLE.**

| Characteristics | n = 58 |
|---|---|
| Gender | |
| Female, n (%) | 50 (86) |
| Male, n (%) | 8 (14) |
| Age, median (range), years | 36 (16–79) |
| Duration of SLE, median (range), month | 18 (0–32) |
| Untreated patients (%) | 52(90) |
| Components of SLE diagnostic criteria, n (%) | |
| Skin rash | 30 (52) |
| Oral ulcers | 4 (7) |
| Alopecia | 1 (2) |
| Arthritis | 20 (34) |
| Serositis | 15 (26) |
| Renal disease | 29(50) |
| CNS disease | 11(19) |
| Hemolytic anemia | 9 (15) |
| Laboratory findings | |
| Leukocytopenia | 21 (36) |
| Thrombocytopenia | 15 (26) |
| Anti-ds-DNA Ab positive | 46 (79) |
| Anti-smith Ab positive | 28 (48) |
| Anti-phospholipid Ab positive | 27 (47) |
| SLEDAI, median(range) | 13(0–50) |
| SDI, median(range) | 1(0–4) |

CNS = central nervous system, Ab = antibody, SLEDAI = SLE Disease Activity, SDI = Systemic Lupus International Collaborating Clinics (SLICC) damage index

correlations with Gal-9. Serum Gal-9 levels were significantly correlated with serum CXCL 10 levels as well as M2BPGi (Fig 3).

We then examined the relationship between serum levels of CXCL 10 or M2BPGi and the SLE disease activity (Fig 4A and Fig 4B). A significant correlation was found between SLE-DAI-2K score and circulating levels of M2BPGi (Fig 4A). Whereas there was no significant correlation between SLEDAI-2K score and circulating levels of CXCL 10 (Fig 4B).

We then divided the patients into groups with or without organ damage measured by the SLICC/ACR Damage Index (SDI) and compared them with the circulating levels of Gal-9 (A), CXCL10 (B) or M2BPGi (C). As shown in Fig 5A, serum levels of Gal-9 were significantly higher in SLE patients with at least one organ damage (SDI $\geq$ 1) compared with those without organ involvement (23.1 ng/ml [IQR: 4.8–59.7] versus 12.5 ng/ml [IQR: 3.6–36.6], $p$ = 0.013). Conversely, there was no significant difference in serum levels of CXCL 10 (240.2 ng/ml [IQR: 20.9–2664.8] versus 299.6 ng/ml [IQR: 0.4–2570.4], $p$ = 0.89) or M2BPGi (2.5 ng/ml [IQR: 0.6–9.5] versus 2.1ng/ml [IQR: 0.2–6.2], $p$ = 0.38) between SLE patients with and those without organ damage (Fig 5B and Fig 5C).

The SDI scores irreversible organ damage that occurred since the onset of SLE regardless of cause. The British Isles lupus assessment Group (BILAG) is used to evaluate the specific manifestations over the previous 4 weeks in a total 8 organ systems. BILAG was found to be useful tool in monitoring disease activity in SLE patients, which was developed to report disease activity in eight organ systems including renal or neurological manifestations [18]. Details of

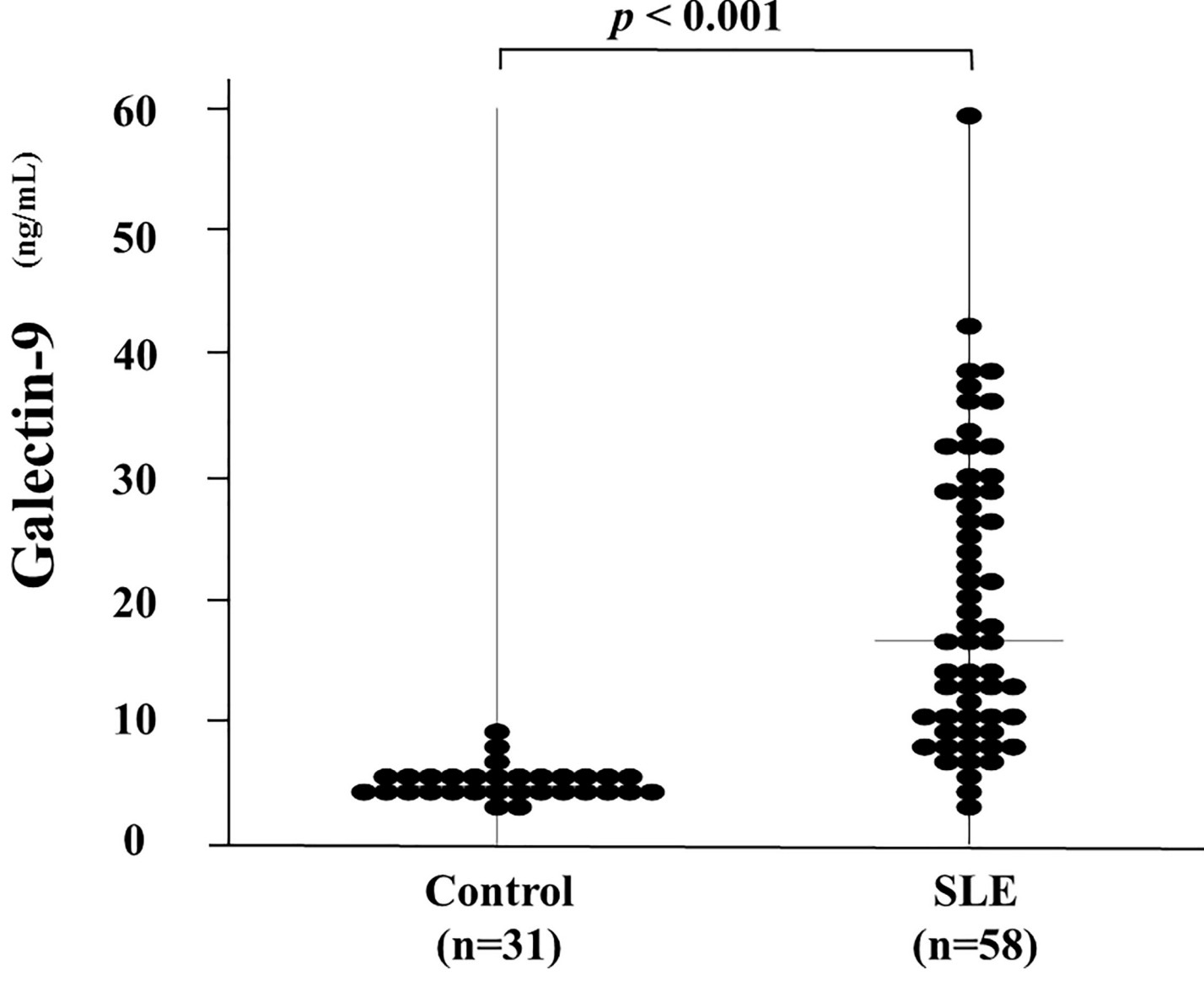

**Fig 1. Serum levels of Gal-9 in SLE patients (n = 58) and healthy subjects (n = 31).** Higher Gal-9 levels were found in patients with SLE compared with those in healthy subjects. Median Gal-9 levels (bar) are depicted and statistical analysis was performed using the Mann-Whitney U test.

patient's distribution according to the results of BILAG scores are shown in Table 2. Among 8 domains of BILAG grading, active organ involvements were detected mainly in neurological, renal and hematological domains. As shown in Fig 6, in SLE patients with active renal involvement (BILAG renal A-B), the serum levels of Gal-9 were significantly higher compared to those without active renal involvement (BILAG renal C-E). Whereas there was no significant difference in serum levels of Gal-9 between SLE patients with (BILAG A-B) or without active neurological or hematological involvements (BILAG C-E).

## Relationships between serum levels of IFN-α and organ involvements

To examine whether serum IFN-α concentrations were relevant to SLE disease activity, we measured IFN-α using the same sera isolated from SLE patients. IFN-α was detectable in the

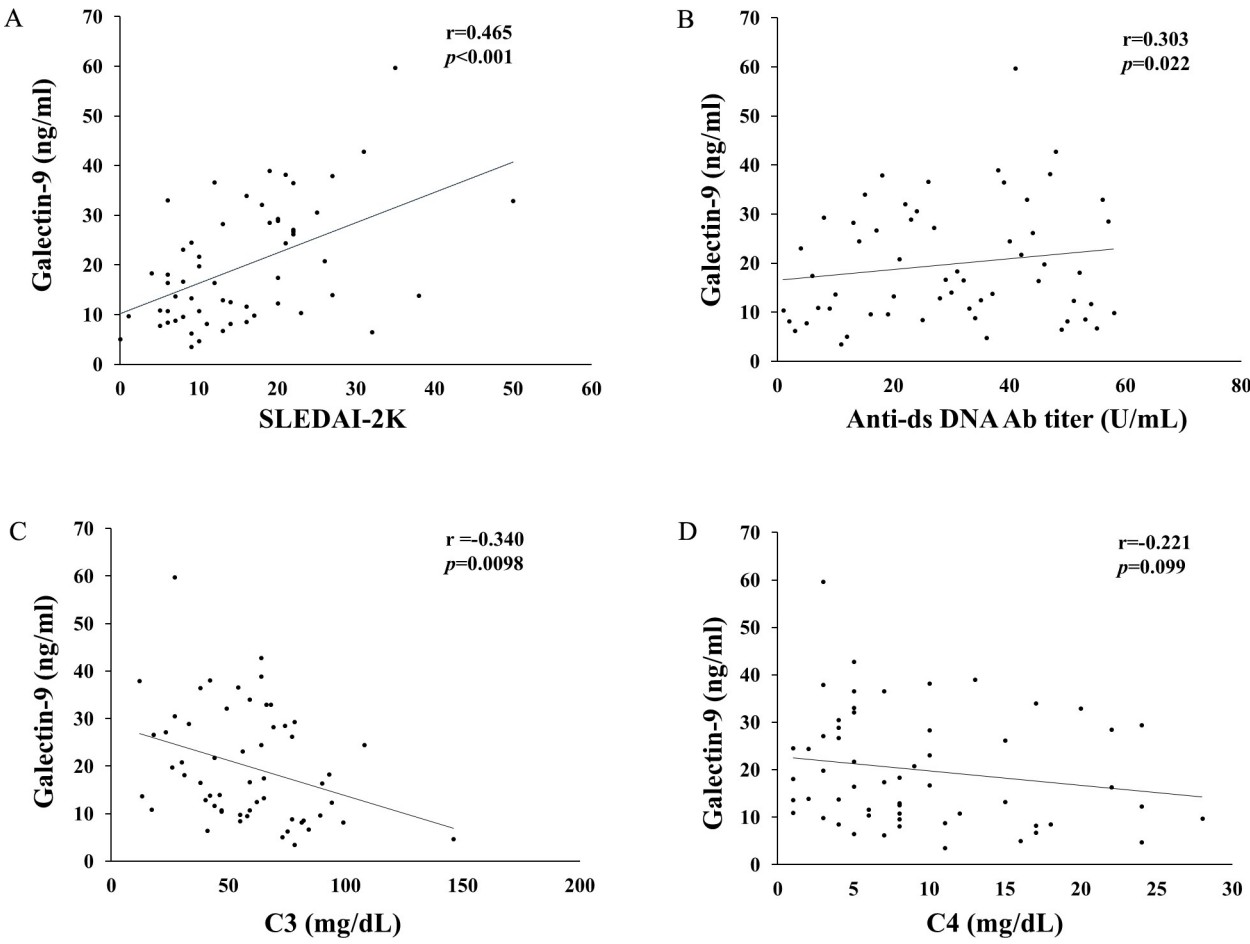

**Fig 2.** Correlations between serum levels of Gal-9 and clinical parameters (A: SLEDAI-2K, B: anti-ds-DNA Ab, C: C3 and D: C4) in SLE patients. Serum levels of Gal-9 significantly positively correlated with SLEDAI-2K (A) or anti-ds-DNA Ab (B) and negatively correlated with C3 levels (C). There was no significant correlation between serum levels of Gal-9 and C4 levels (D). Statistics and regression line are represented by the solid line.

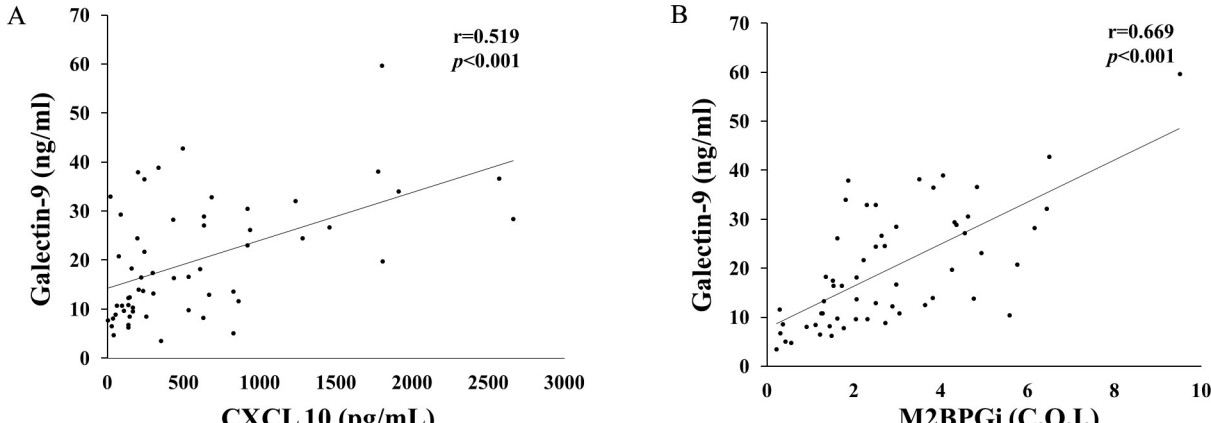

**Fig 3.** Correlations between serum levels of Gal-9 and CXCL 10 (A) or M2BPGi (B) in patients with SLE. Serum levels of Gal-9 significantly correlated with serum levels of CXCL 10 (A) or M2BPGi (B) in patients with SLE. Statistics and regression line are represented by the solid line. Gal-9 = galetin-9, I CXCL10 = C-X-C motif chemokine 10, M2BPGi = Mac-2 binding protein glycosylation isomer.

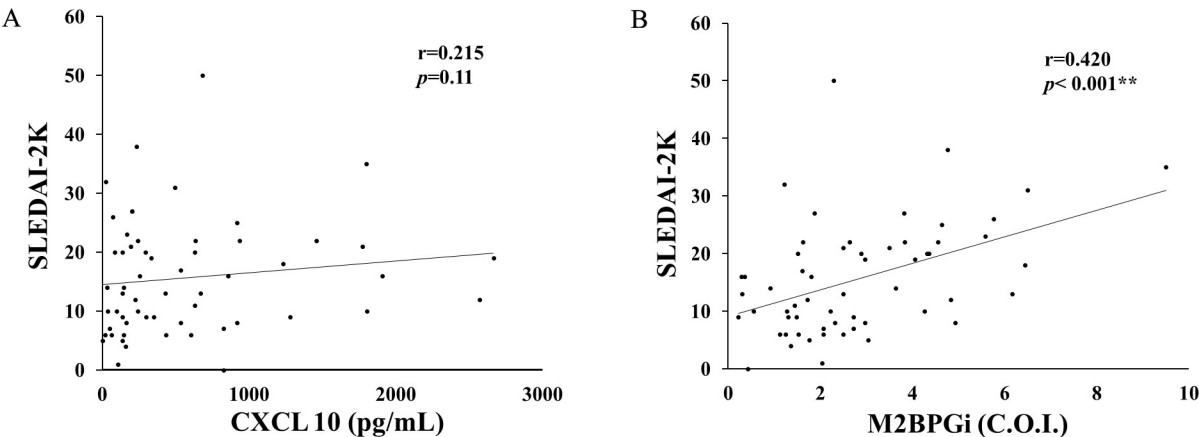

**Fig 4.** 10 (A) or M2BPGi (B) and SLEDAI-2K in patients with SLE. There was no significant correlation between serum levels of Gal-9 and CXCL 10 (A). Serum levels of M2BPGi (B) significantly correlated with SLEDAI-2K in SLE patients. Statistics and regression line are represented by the solid line. Gal-9 = galetin-9, CXCL10 = C-X-C motif chemokine 10, M2BPGi = Mac-2 binding protein glycosylation isomer.

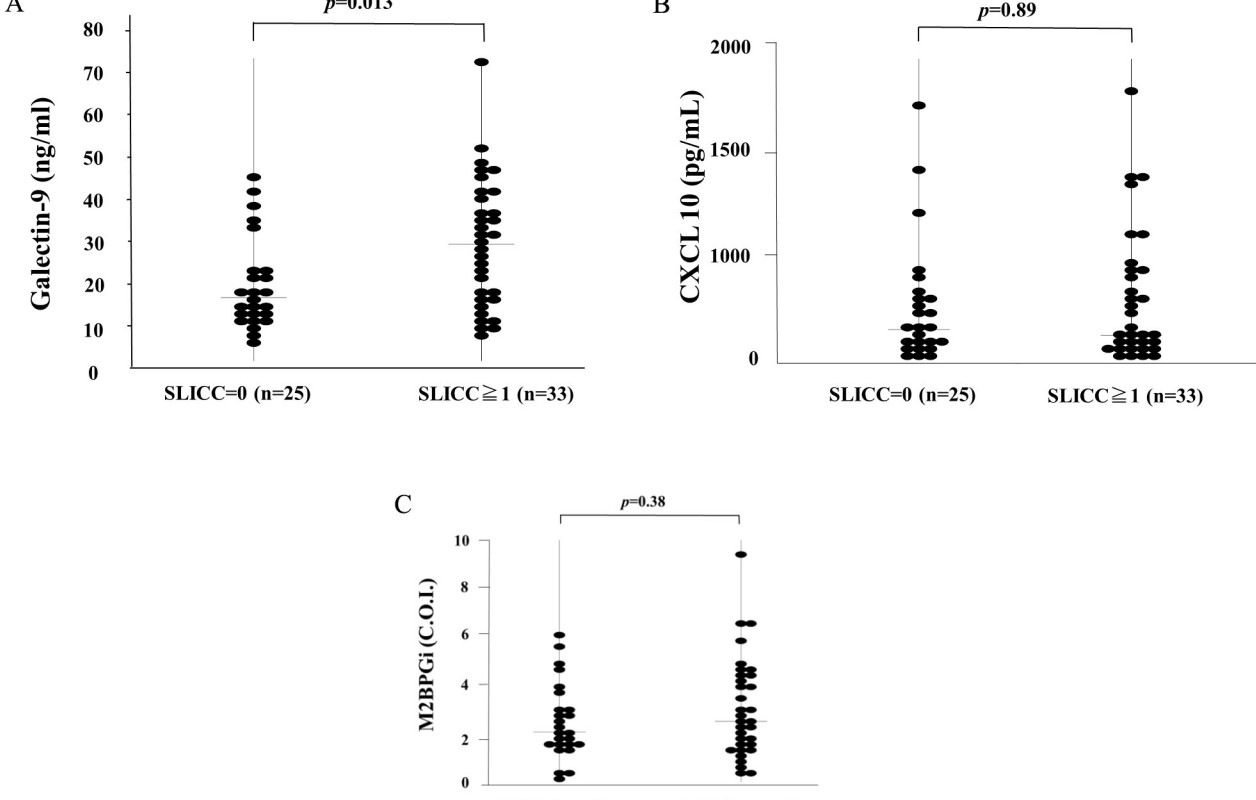

**Fig 5. Serum biomarkers in SLE patients with or without organ damage.** We compared serum levels of biomarkers (A: Gal-9, B: CXCL 10, C: M2BPGi) between SLE patients with any organ damage (i.e., SDI ≥ 1) and those without organ damage (i.e., SDI = 0). Raised serum Gal-9 levels were found in SLE patients with any organ damage compared with those without organ damage (A). No significant difference in CXCL 10 (B) or M2BPGi (C) levels was observed between SLE patients with and without any organ damage. SDI: Systemic Lupus International Collaborating Clinics/American College of Rheumatology Damage Index.

**Table 2. Disease activity as assessed by the British Isles Lupus Assessment Group (BILAG) grading.**

| BILAG | BILAG grade | | | | |
|---|---|---|---|---|---|
| Manifestations | A | B | C | D | E |
| General | 0 | 12 | 14 | 4 | 28 |
| Mucocutaneous | 1 | 4 | 19 | 2 | 32 |
| Neurological | 2 | 10 | 1 | 0 | 45 |
| Musculoskeletal | 1 | 4 | 14 | 2 | 37 |
| Cardiorespiratory | 0 | 6 | 9 | 3 | 40 |
| Gastrointestinal | 0 | 7 | 0 | 0 | 51 |
| Renal | 10 | 5 | 6 | 3 | 34 |
| Hematological | 1 | 15 | 32 | 1 | 9 |
| Opthalmic | 1 | 4 | 6 | 1 | 46 |

BILAG: British Isles Lupus Assessment Group. BILAG grades: *A*:severe, *B*: intermediate, *C*: mild, *D*: inactive, *E*: no activity.

sera of only a subset of SLE patients (21/58, 36.2%, 70.0pg/ml [IQR: 1.4–601.0]). We compared the circulating levels of Gal-9 between SLE patients with detectable serum IFN-α (cut-off value = 1.0pg/ml, n = 21) and those without detectable serum IFN-α (n = 37). As shown in Fig 7, serum levels of Gal-9 were significantly higher in SLE patients with detectable IFN-α compared with those without detectable IFN-α (26.7 ng/ml [IQR: 6.5–59.7] versus 12.5 ng/ml [IQR: 3.6–39.0], $p < 0.002$).

## Longitudinal changes in Gal-9 serum concentrations

To explore the longitudinal changes in Gal-9 and associations with disease activity, we included 5 patients with two longitudinal samples (at least 1 month apart) and with high disease activity. In the longitudinal study, 5 active SLE patients were followed until they became inactive and then resampled (SLEDAI-2K before 23.4± 7.6, after 6.0 ± 5.1). The median duration of follow-up was 2 months (1–12 months). The levels of serum Gal-9 decreased significantly after treatment with glucocorticoids and immunosuppressive drugs (Fig 8). Therefore, serum Gal-9 levels in patients with active SLE were diminished following successful treatment with clinical improvement.

## Neuropsychiatric manifestations and CSF levels of Gal-9

Although there was no differences in serum levels of Gal-9 between SLE patients with and without active neurological involvement (BILAG neurological A-B), we determined whether Gal-9 in CSF could be influenced by the presence of NPSLE. To test this hypothesis, we assayed the Gal-9 levels in CSF isolated from 18 patients with NPSLE. As shown in Fig 9, levels of Gal-9 in CSF samples from patients with neuropsychiatric involvement (NPSLE) were significantly higher (6.7 ±7 .4 ng/ml, media [IQR: 1.0–27.2]), $p = 0.0093$) compared to non-SLE controls (1.3 ± 0.4 ng/ml, media [IQR: 0.9–2.1]).

Although the paired samples (CSF plus serum) were available in a limited number of subjects, we compared CSF Gal-9 with serum Gal-9 levels in patients with NPSLE (n = 11) or controls (n = 4). The relationships between CSF Gal-9 and serum Gal-9 were shown in Fig 10. It is likely that CSF Gal-9 levels seem to be elevated regardless of the values of serum Gal-9 levels in a subset of NPSLE patients.

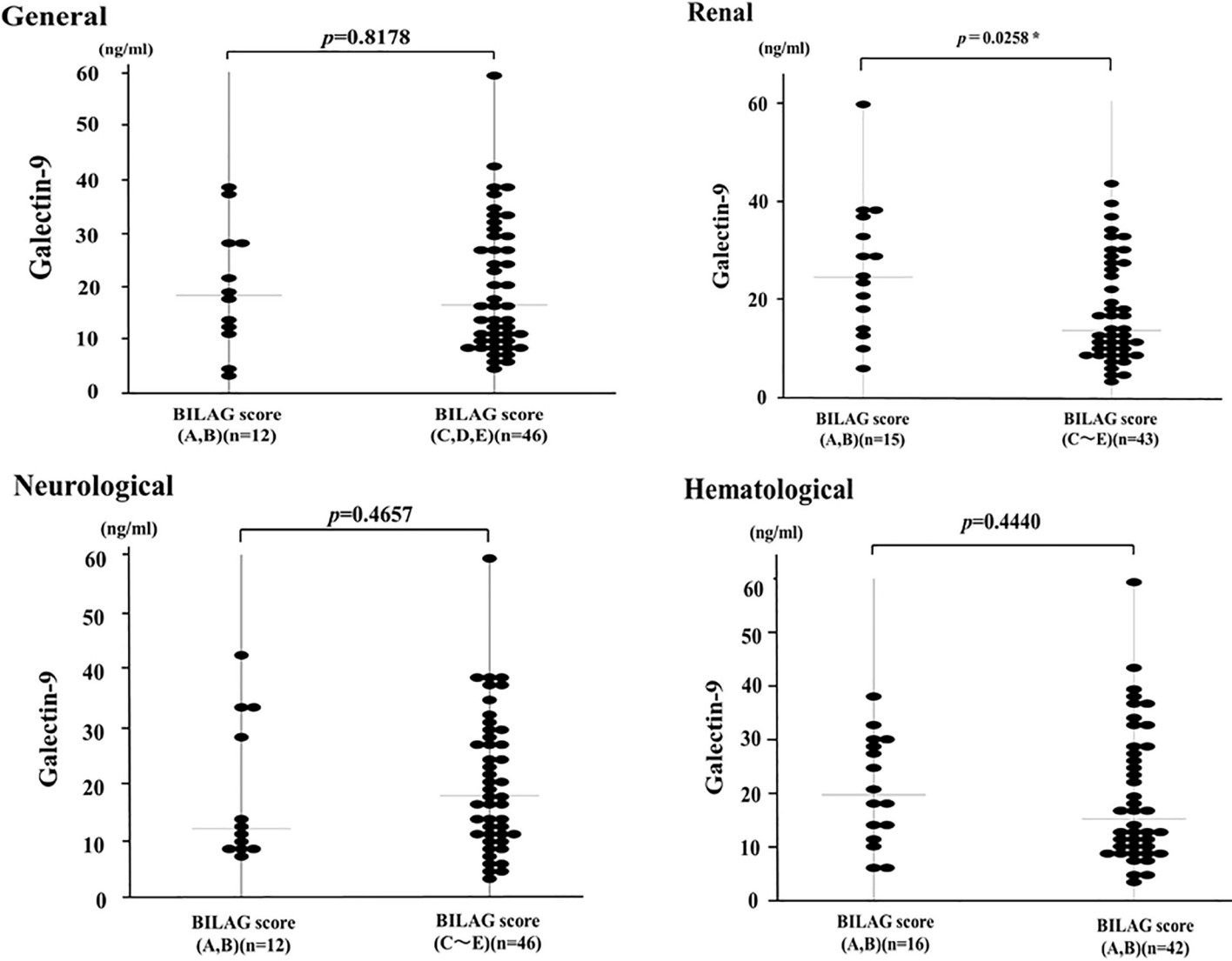

**Fig 6. Serum levels of Gal-9 in SLE patients with or without active organ involvements.** We compared serum levels of Gal-9 between SLE patients with active organ involvements (BILAG general, renal, neurological, hematological domains; A-B) and without active organ involvement (BILAG; C-E).

## Discussion

SLE is an autoimmune disease characterized by systemic vasculitis and inflammation of connective tissues leading to multiple organ damage [3]. Many studies have been performed to identify reliable biomarkers for SLE including non-invasive and easily measurable surrogates which are able to assess disease activity or treatment response [22]. Several cytokines, particularly type I IFNs, are implicated in the pathogenesis of SLE [4]. Despite an important role of type 1 IFNs, the direct quantification of type 1 IFNs has been challenging. We focused on Galectin-9, which was shown to be a potential biomarker for the interferon signature [7], with regard to SLE disease activity. The major finding in this study is that circulating levels of Gal-9 are elevated in patients with SLE and are correlated with SLE disease activity and could be a discriminator between SLE patients with and without organ damage distinguished by SLCC

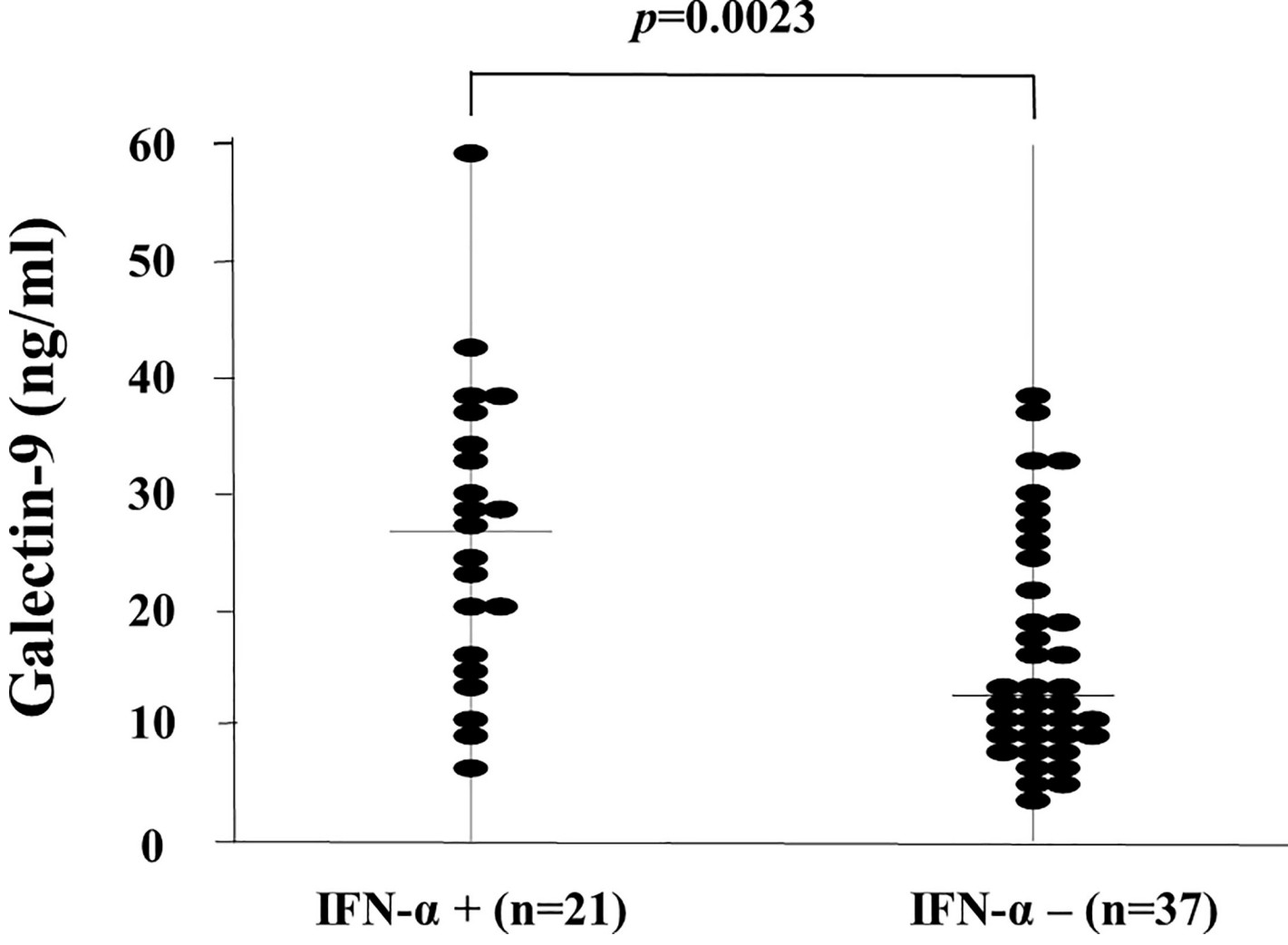

**Fig 7. Serum Galectin-9 (Gal-9) levels in SLE patients with detectable circulating IFN-α.** We compared serum levels of Gal-9 between SLE patients with and those without detectable circulating IFN-α. Raised serum Gal-9 levels were found in SLE patients with detectable circulating IFN-α compared with those without detectable circulating IFN-α. Median Gal-9 levels (bar) are depicted and statistical analysis was performed using the Mann-Whitney U test.

damage score. Our results suggest that Gal-9 could be a predictor for SLE disease phenotype and deserves attention as a clinically useful biomarker.

Tim-3 has been implicated in the pathogenesis of autoimmune diseases [23]. Gal-9 is one of the β-galactosidase binding lectins that suppresses Th1 cell or Th17 cells in a Tim-3-dependent or independent manner [24]. SLE patients had been demonstrated to contain high proportions of CD3$^+$CD4$^+$Tim-3$^+$ T cell subsets compared with healthy subjects and Tim-3 expression on T cells correlates with SLE disease activity [25]. Galectin-9 is ubiquitously expressed in a variety of tissues, whereas, Galectin-9 expressed mainly in the immune system [26]. Galectin-9 expression is induced by IFN-γ suggesting a feedback mechanism whereby the IFN-γ induces tissue inflammation also induces an inhibitory ligand, Galectin-9 [27], which suppresses the differentiation of Th1/Th17 cells in Tim-3-dependent manner [28]. These close associations between Tim-3 expression and SLE disease activity could be linked with elevated levels of circulating Gal-9, a specific ligand of Tim-3. Our findings suggest that the circulating levels of

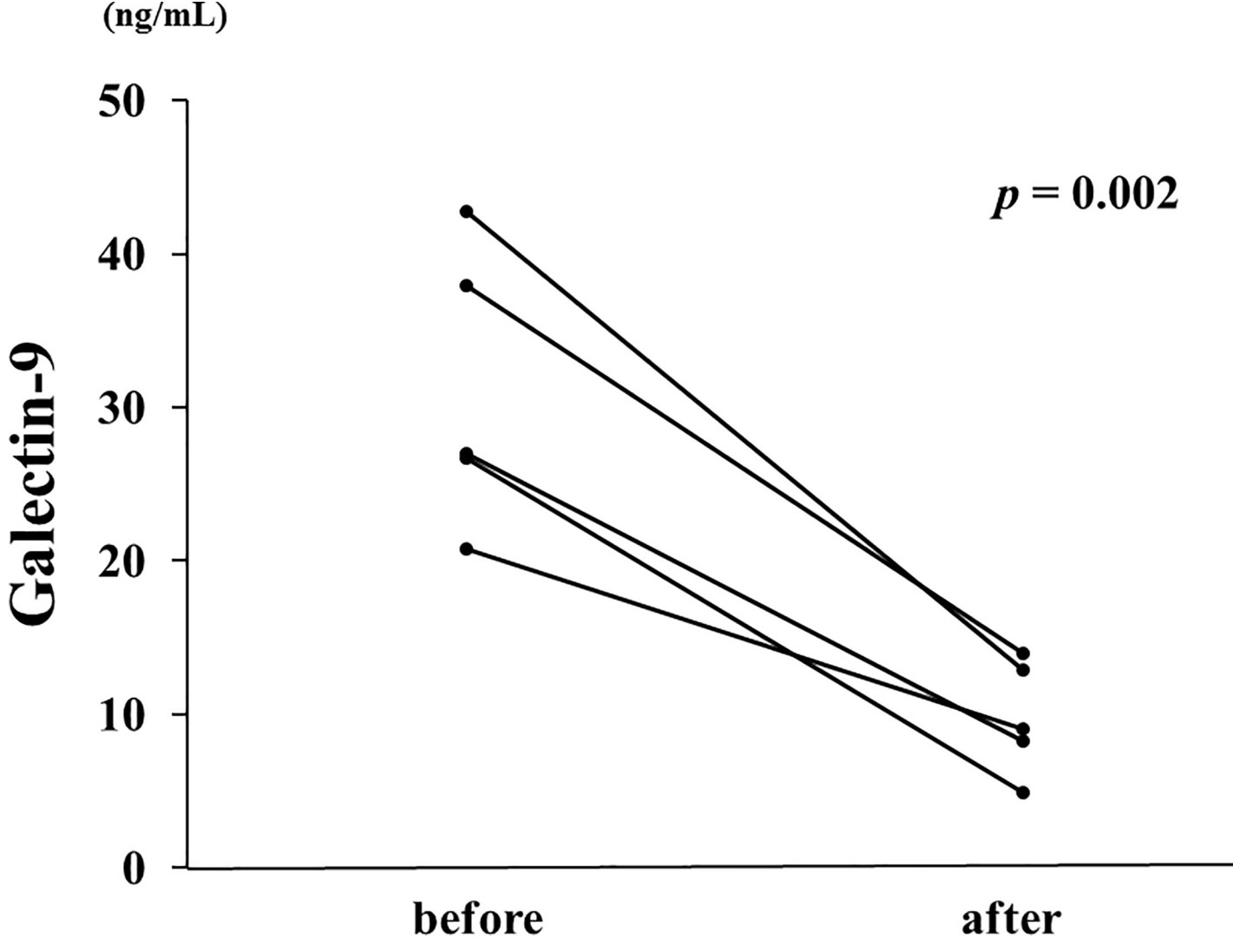

**Fig 8. Longitudinal changes in serum Gal-9 concentrations in 5 patients with active SLE before and after immunosuppressive treatments.** Paired samples from the same subjects were compared by Wilcoxon signed-rank test. Gal-9 = galetin-9.

Gal-9 may be useful to evaluate the SLE disease activity and highlight Gal-9 as a potential bio-marker for SLE.

In this study we also demonstrated that Gal-9 is elevated in CSF isolated from patients with NPSLE. NPSLE involving the central nervous system is a life-treating manifestation of SLE [29]. Although the pathophysiology of NPSLE is not completely determined, research into the underlying mechanisms has focused on autoantibodies or cytokines [30]. It has been reported that certain autoantibodies or cytokines are relevant to NPSLE [31]. CSF studies have been instrumental in evaluating immunobiomarkers related to the immune dysfunction in NPSLE [32]. NPSLE is often associated with the presence of neuropathic cytokines such as IL-6 and IFN-α in CSF [33]. Additionally, the passage of autoantibodies across the blood-brain barrier (BBB) had been suggested in NPSLE [34]. To our knowledge, CSF levels of Gal-9 in NPSLE had never been described and no study has shown involvement of Gal-9 in NPSLE. The present study showed that elevated levels of Gal-9 in CSF were observed in patients with NPSLE

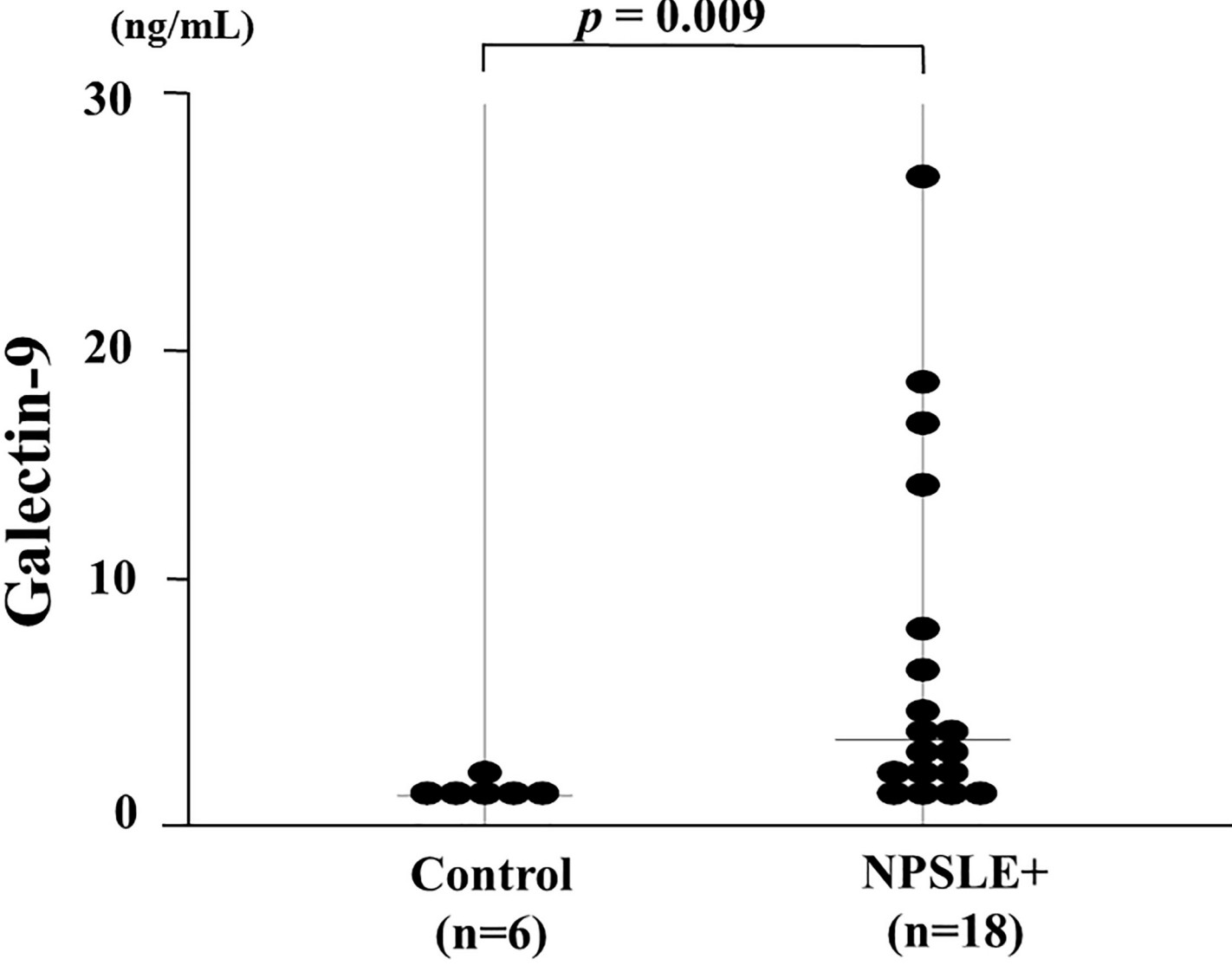

**Fig 9. Levels of Galectin-9 (Gal-9) in cerebrospinal fluid (CSF).** Samples of CSF were obtained from patients with neuropsychiatric systemic lupus erythematosus (NPSLE, n = 18) and non-SLE controls (n = 6). Each point represents an individual patient. Median Gal-9 levels (bar) are depicted and statistical analysis was performed using the Mann-Whitney U test.

and is the first to show the clinical significance of CSF Gal-9. These findings suggest that Gal-9 might also play a pathogenic role in NPSLE.

Our study demonstrated that Gal-9 levels are correlated with serum levels of CXCL-10 or M2BPGi. Furthermore, M2BPGi is correlated with SLE disease activity. We suggest that M2BPGi could be used in clinics as a noninvasive and useful biomarkers for SLE activity as described previously [14]. Serum Gal-9 levels were significantly higher in patients with SLE-related organ involvement than in SLE patients without organ involvement. Conversely, there was no significant difference in circulating CXCL10 and M2BPGi between SLE patients with and without organ damage. These findings suggest that only Gal-9 could be useful biomarker that reflects SLE organ involvements among these biomarkers. Analysis using longitudinal samples from active SLE patients also demonstrated that serum levels of Gal-9 are useful to assess the therapeutic changes of SLE disease activity. It has been generally accepted that high

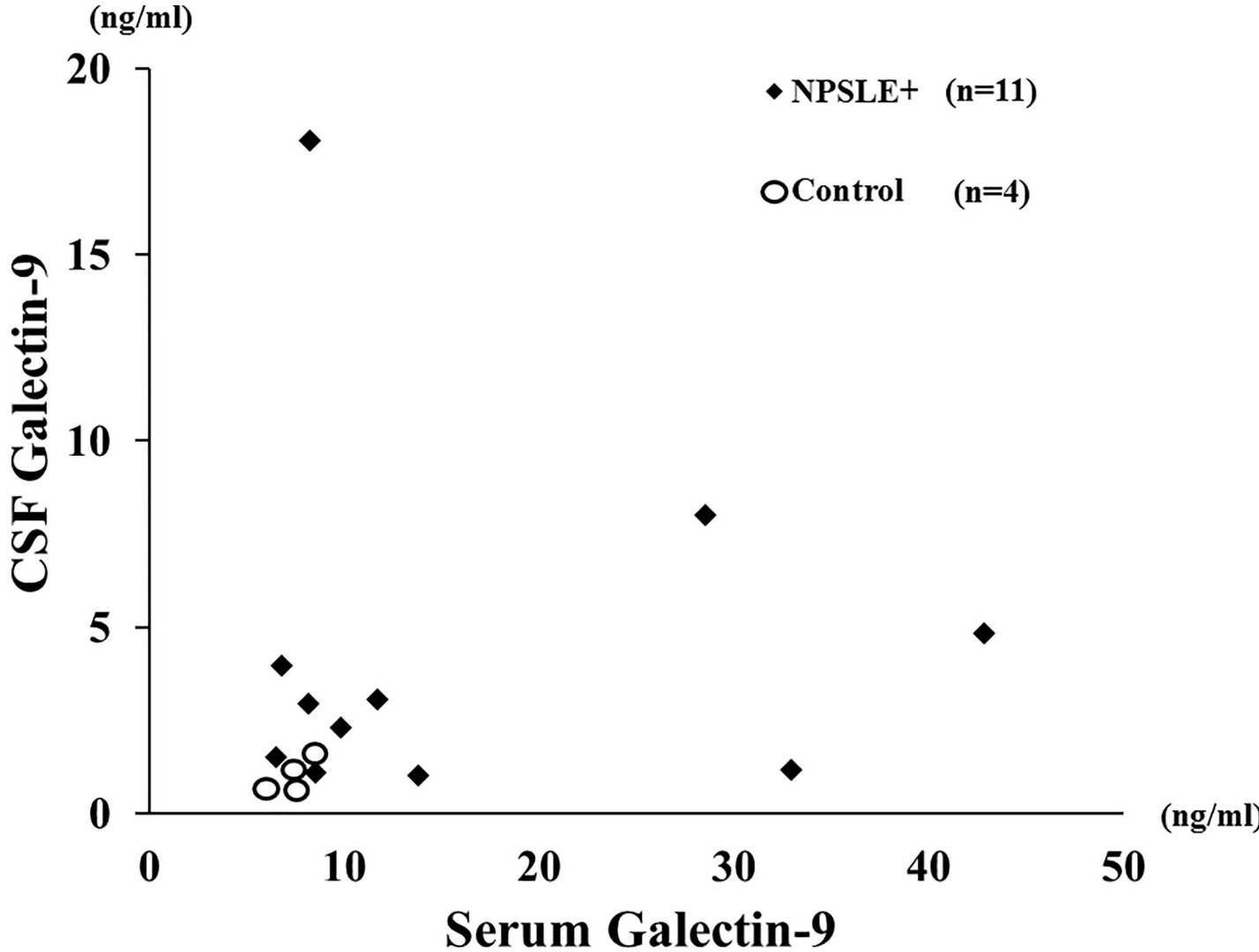

**Fig 10. Relationship between cerebrospinal fluid (CSF) Gal-9 and serum Gal-9 in NPSLE patients and controls.** Relationship between cerebrospinal fluid (CSF) Gal-9 and serum Gal-9 were evaluated in patients with NPSLE and non-SLE controls. NPSLE: neuropsychiatric systemic lupus erythematosus.

disease activity increase the risk of subsequent organ damage in SLE [35]. Indeed, the link between SLE disease activity and organ damage has been demonstrated in recent studies. Lopez et al demonstrated that SLE disease activity measured by BILAG score predicts the risk of subsequent organ damage and mortality [36]. Although serum levels of Gal-9 were demonstrated to be a promising marker to assess the IFN signature, our data suggest that Gal-9 could be a biomarker for SLE disease activity or particular SLE-related organ involvement. Elevated levels of Gal-9 were demonstrated in patients with primary Sjögren's syndrome, and to be correlated with disease activity assessed by EULAR Sjögren's syndrome Disease Activity Index (ESSDAI) [37]. Additionally, Gal-9 was validated as reliable biomarker for disease activity in juvenile dermatomyositis (DM) [38]. In accordance with these reports, our data suggest that Gal-9 could be a biomarker that reflects the particular SLE-related organ involvement in addition to SLE disease activity.

This study has some limitations. First, the sample size is relatively small and the design is cross-sectional. Second the CSF samples were not obtained from SLE patients without NPSLE.

Third, the methods for identifying type I IFN signatures such as the global gene expression analysis were not performed in this study. Finally, our sample is mainly composed of patients with recent-onset SLE, we recognize that this aspect might have impact on the organ damage due to the short follow-up periods in contrast to other cohorts with different characteristics. A more detailed association of Gal-9 with lupus-related organ damage should be elucidated in a large-scale study.

## Conclusions

In conclusion, our study demonstrates that serum Gal-9 has clinical association in SLE, in particular highlighting the associations between Gal-9 and SLE disease activity and organ involvement. These data suggest that Gal-9 may be linked with SLE-mediated organ involvements. Further research is required to elucidate the importance of Gal-9 in SLE, including NPSLE to determine how Gal-9 might be modulated by SLE phenotype.

## Acknowledgments

We are grateful to Ms Kanno Sayaka for her technical assistance in this study.

## Author Contributions

**Conceptualization:** Naoki Matsuoka, Yuya Fujita, Jumpei Temmoku, Makiko Yashiro Furuya, Tomoyuki Asano, Shuzo Sato, Haruki Matsumoto.

**Data curation:** Naoki Matsuoka, Yuya Fujita, Haruki Matsumoto, Hiroko Kobayashi, Eiji Suzuki.

**Formal analysis:** Naoki Matsuoka, Hideko Kozuru, Kiyoshi Migita.

**Investigation:** Naoki Matsuoka, Hideko Kozuru, Kiyoshi Migita.

**Methodology:** Hideko Kozuru, Kiyoshi Migita.

**Project administration:** Hiroshi Watanabe, Kiyoshi Migita.

**Resources:** Naoki Matsuoka, Kiyoshi Migita.

**Supervision:** Hiroshi Yastuhashi, Kiyoshi Migita.

**Validation:** Naoki Matsuoka, Kiyoshi Migita.

**Writing – original draft:** Naoki Matsuoka, Kiyoshi Migita.

**Writing – review & editing:** Naoki Matsuoka, Shuzo Sato, Kiyoshi Migita.

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
