## [Decision Letter · Decision Letter 0]

15 Oct 2019

PONE-D-19-27302

Galectin-9 as a biomarker for disease activity in systemic lupus erythematosus

PLOS ONE

Dear Dr. Migita,

Thank you for submitting your manuscript to PLOS ONE. After careful consideration, we feel that it has merit but does not fully meet PLOS ONE’s publication criteria as it currently stands. Therefore, we invite you to submit a revised version of the manuscript that addresses the points raised during the review process.

Both reviewers found some interests in this article, but there are a number of points that require improvement, or oven amendment. Especially, comments made by reviewer #1 is critical. Submission of revised version requires thorough responses to all comments made by reviewers, and massive rewriting the manuscript.

We would appreciate receiving your revised manuscript by Nov 29 2019 11:59PM. To enhance the reproducibility of your results, we recommend that if applicable you deposit your laboratory protocols in protocols.io, where a protocol can be assigned its own identifier (DOI) such that it can be cited independently in the future. For instructions see: http://journals.plos.org/plosone/s/submission-guidelines#loc-laboratory-protocols

We look forward to receiving your revised manuscript.

Kind regards,

Masataka Kuwana, MD, PhD

Academic Editor

PLOS ONE

Journal Requirements:

'The study was supported by the Practical Research Project for Rare / Intractable Diseases from Japan Agency for Medical Research and Development, AMED.'

3. Please include your tables as part of your main manuscript and remove the individual files. Please note that supplementary tables (should remain/ be uploaded) as separate "supporting information" files

Additional Editor Comments (if provided):

Reviewers' comments:

Reviewer's Responses to Questions

**Comments to the Author**

1. Is the manuscript technically sound, and do the data support the conclusions?

Reviewer #1: Partly

Reviewer #2: Yes

2. Has the statistical analysis been performed appropriately and rigorously? 

Reviewer #1: No

Reviewer #2: Yes

3. Have the authors made all data underlying the findings in their manuscript fully available?

Reviewer #1: Yes

Reviewer #2: Yes

4. Is the manuscript presented in an intelligible fashion and written in standard English?

Reviewer #1: No

Reviewer #2: Yes

5. Review Comments to the Author

Reviewer #1: First of all, although the authors claimed that Galectin-9 (Gal-9) could be a serologic marker of disease activity and organ damage in systemic lupus erythematosus (SLE) patients, a single biomarker is rarely associated with SLE disease activity and damage concomitantly because damage in SLE patients is irreversible change in organs usually due to longstanding disease activity. Thus, that would rather raise suspicion over the accuracy of the SLEDAI-2K scores and SLICC damage index in the present study. Alternatively, the true association between Gal-9 and SLE. This issue should be clarified in the manuscript.

In addition, although the authors reported that Gal-9 is elevated in CSF from patients with NPSLE and claimed that the present study showed the clinical significance of CSF Gal-9, most of the previously reported candidate CSF biomarkers for NPSLE were elevated exclusively in CSF but not in serum. Thus, that would rather raise suspicion over the significance of Gal-9 as a biomarker for SLE. This issue should be clarified in the manuscript, too.

In the abstract, “damage scores” should be written as the SLICC damage index.

In the abstract, values such as “16.6 ng/ml [3.6-59.5]” should be explained that they were medians and 5–95th centiles.

In the abstract, although the authors wrote that levels of Gal-9 were also significantly higher in the CSF from patients with recent-onset neuropsychiatric SLE (NPSLE) than in those from non-SLE controls, the representative values and statistics should be mentioned.

In the Methods section, original papers should be cited for SLEDAI-2K and SLICC damage index instead of review articles.

In the Submission Guidelines of this journal, it is stated that all research involving human participants must have been conducted according to the principles expressed in the Declaration of Helsinki. This should probably be mentioned in the manuscript.

In the Methods section, it was stated that 7 patients with NPSLE were included outside of the enrolled 58 SLE paints in whom serum samples were not collected. These 7 NPLSE patients should be included in the study population. Moreover, it is difficult to understand why these 7 NPLSE patients were excluded from the serum analyses.

In the Methods section, it was stated that results were non-normally distributed and were compared by the Mann-Whitney U test. Then why correlations between continuous variables were analyzed by the Pearson correlation test?.

In the Results section, p values were reported for correlation analyses. However, the correlation is independent of sample size, whereas the p value is affected by sample size. Thus, correlation coefficients and their confidence intervals are more important than p values.

In the Discussion section, the first several sentences (from “SLE is a systemic autoimmune condition” to “SLE disease activity and organ involvements” should be discussed in the Introduction section.

In the Discussion section of the manuscript, the authors claimed that Gal-9 is a more powerful discriminator between SLE patients with organ damage compared with CXCL10 and M2BPGi. If the authors believe this, ROC curve analyses should be performed and the sensitivity and specificity for the thresholds should be demonstrated. From the dot plots where many values are in the same ranges, it does not seem that serum levels of Gal-9 are able to discriminate SLE patients with and without organ damage, though.

Throughout the manuscripts, there are grammatical errors.

Reviewer #2: Authors demonstrated that galectin-9 is a serological marker of disease activity and organ damage in SLE patients. The new correlation between galectin-9 titer and organ damage is different from the previous paper (ref # 7).

1)Discuss why galectin-9 increases with organ damage from the perspective of cells that produce galectin-9.

2)Galectin-9 is also a biomarker in other autoimmune diseases, such as primary sjogren syndrome (ARD in press, 2018-214651) and juvenile dermatomyositis (A & R 71 (8): 1377-90, 2019) and correlates with disease activity in each disease. It is not a marker specific to SLE. Please discuss from this point of view.

6. PLOS authors have the option to publish the peer review history of their article (what does this mean?). If published, this will include your full peer review and any attached files.

Reviewer #1: No

Reviewer #2: No

---

## [Author Response · Author response to Decision Letter 0]

25 Nov 2019

Reviewer #1: First of all, although the authors claimed that Galectin-9 (Gal-9) could be a serologic marker of disease activity and organ damage in systemic lupus erythematosus (SLE) patients, a single biomarker is rarely associated with SLE disease activity and damage concomitantly because damage in SLE patients is irreversible change in organs usually due to longstanding disease activity. Thus, that would rather raise suspicion over theaccuracy of the SLEDAI-2K scores and SLICC damage index in the present study. Alternatively, the true association between Gal-9 and SLE. This issue should be clarified in the manuscript.

We appreciate your critical comments. The SDI scores irreversible organ damage that occurred since the onset of SLE regardless of cause. The British Isles lupus assessment Group (BILAG) is used to evaluate the specific manifestations and to be useful tool in monitoring disease activity in SLE patients. According to your critical comments, we evaluated the disease activity and SLE-related organ involvement using this BILAG sore. We presented these new data and discuss this important point in the revised manuscript.

In addition, although the authors reported that Gal-9 is elevated in CSF from patients with NPSLE and claimed that the present study showed the clinical significance of CSF Gal-9, most of the previously reported candidate CSF biomarkers for NPSLE were elevated exclusively in CSF but not in serum. Thus, that would rather raise suspicion over the 

significance of Gal-9 as a biomarker for SLE. This issue should be clarified in the manuscript, too.

We appreciate your critical comments. According to your important comments, we compared CSF Gal-9 with serum Gal-9 in patients with NPSLE and controls. As pointed out in your comment, our data suggested that CSF Gal-9 levels seems to be elevated regardless of the values of serum Gal-9 levels in a subset of NPSLE patients. We added these data in the revised manuscript. 

In the abstract, “damage scores” should be written as the SLICC damage index.

In the abstract, values such as “16.6 ng/ml [3.6-59.5]” should be explained that they were medians and 5–95th centiles.

According to your precise comments, we corrected and described the range of the presented data as interquartile range (IQR).

In the abstract, although the authors wrote that levels of Gal-9 were also significantly higher in the CSF from patients with recent-onset neuropsychiatric SLE (NPSLE) than in those from non-SLE controls, the representative values and statistics should be mentioned.

According to your precise comments, we added these data in the Abstract. 

In the Methods section, original papers should be cited for SLEDAI-2Kand SLICC damage index instead of review articles.

We appreciate your critical comments. We cited the proper reference forSLEDAI-2K and SLICC damage index in the revised manuscript.

In the Submission Guidelines of this journal, it is stated that all research involving human participants must have been conducted according to the principles expressed in the Declaration of Helsinki. This should probably be mentioned in the manuscript.

We appreciate your critical comments. We mentioned this important issues including ethnical approval in the revised manuscript.

In the Methods section, it was stated that 7 patients with NPSLE were included outside of the enrolled 58 SLE paints in whom serum samples were not collected. These 7 NPLSE patients should be included in the study population. Moreover, it is difficult to understand why these 7 NPLSE patients were excluded from the serum analyses.

We appreciate your critical comments. Due to the small number of the paired samples, (CSF plus sera), we could not analyze the CSF plus sera in a sufficient number of patients.

In the Methods section, it was stated thatresults were non-normally distributed and were compared by the Mann-Whitney U test. Then why correlations between continuous variables were analyzed by the Pearson correlation test?.

In the Results section, p values were reported for correlation analyses. 

However, the correlation is independent of sample size, whereas the p value is affected by sample size. Thus, correlation coefficients and their confidence intervals are more important than p values.

We appreciate your critical comments. According to your precise comments, we evaluated the corrections between Gal-9 and clinical parameters using Spearman’s correction test. We also calculated the confidence intervals. We corrected these descriptions and added these data in revised manuscript. 

In the Discussion section, the first several sentences (from “SLE is a systemic autoimmune condition” to “SLE disease activity and organ involvements” should be discussed in the Introduction section.

We appreciate your precise comments. According to your comments, we revised these description and discuss this important point in the revised manuscript.

In the Discussion section of the manuscript, the authors claimed that Gal-9 is a more powerful discriminator between SLE patients with organ damage compared with CXCL10 and M2BPGi. If the authors believe this, ROC curve analyses should be performed and the sensitivity andspecificity for the thresholds should be demonstrated. From the dot plots where many values are in the same ranges, it does not seem that serum levels of Gal-9 are able to discriminate SLE patients with and without organ damage, 

though.

We appreciate your critical comments. According to your comments, it is difficult to discriminate SLE patients with or without organ damage using this single biomarker (Galectin-9), therefore, we deleted these descriptions in the revised manuscript.

Throughout the manuscripts, there are grammatical errors.

Reviewer #2: Authors demonstrated that galectin-9 is a serological marker of disease activity and organ damage in SLE patients. The newcorrelation between galectin-9 titer and organ damage is different from the previous paper (ref # 7).

1)Discuss why galectin-9 increases with organ damage from the perspective of cells that produce galectin-9.

We appreciate your critical comments. According to your comments, we discussed this important point in the revised manuscript.

2)Galectin-9 is also a biomarker in other autoimmune diseases, such as primary sjogren syndrome (ARD in press, 2018-214651) and juvenile dermatomyositis (A & R 71 (8): 1377-90, 2019) and correlates with disease activity in each disease. It is not a marker specific to SLE. Please discuss from this point of view.

We appreciate your critical comments. According to your precise comments, we cited these important references and discussed this point of view in the revised manuscript. 

----

---

## [Decision Letter · Decision Letter 1]

12 Dec 2019

Galectin-9 as a biomarker for disease activity in systemic lupus erythematosus

PONE-D-19-27302R1

Dear Dr. Migita,

We are pleased to inform you that your manuscript has been judged scientifically suitable for publication and will be formally accepted for publication once it complies with all outstanding technical requirements.

With kind regards,

Masataka Kuwana, MD, PhD

Academic Editor

PLOS ONE

Additional Editor Comments (optional):

Reviewers' comments:

Reviewer's Responses to Questions

**Comments to the Author**

1. If the authors have adequately addressed your comments raised in a previous round of review and you feel that this manuscript is now acceptable for publication, you may indicate that here to bypass the “Comments to the Author” section, enter your conflict of interest statement in the “Confidential to Editor” section, and submit your "Accept" recommendation.

Reviewer #2: (No Response)

2. Is the manuscript technically sound, and do the data support the conclusions?

Reviewer #2: Yes

3. Has the statistical analysis been performed appropriately and rigorously? 

Reviewer #2: Yes

4. Have the authors made all data underlying the findings in their manuscript fully available?

Reviewer #2: Yes

5. Is the manuscript presented in an intelligible fashion and written in standard English?

Reviewer #2: Yes

6. Review Comments to the Author

Reviewer #2: (No Response)

7. PLOS authors have the option to publish the peer review history of their article (what does this mean?). If published, this will include your full peer review and any attached files.

Reviewer #2: No

---

## [Editor Report · Acceptance letter]

6 Jan 2020

PONE-D-19-27302R1 

Galectin-9 as a biomarker for disease activity in systemic lupus erythematosus 

Dear Dr. Migita:

I am pleased to inform you that your manuscript has been deemed suitable for publication in PLOS ONE. Congratulations! Your manuscript is now with our production department. 

With kind regards,

on behalf of

Prof. Masataka Kuwana 

Academic Editor

PLOS ONE